# Healthcare Workers Training Courses on Vaccinations: A Flexible Format Easily Adaptable to Different Healthcare Settings

**DOI:** 10.3390/vaccines8030514

**Published:** 2020-09-08

**Authors:** Laura Serino, Massimo Maurici, Gian Loreto D’Alò, Fabiana Amadori, Elisa Terracciano, Laura Zaratti, Elisabetta Franco

**Affiliations:** 1Department of Biomedicine and Prevention, University of Rome Tor Vergata, 00133 Rome, Italy; lauraserino78@gmail.com (L.S.); maurici@med.uniroma2.it (M.M.); gianloretod@gmail.com (G.L.D.); laura.zaratti@uniroma2.it (L.Z.); 2Department of Prevention, Local Health Unit Roma 1, 00173 Rome, Italy; 3Specialization School for Hygiene and Preventive Medicine, University of Rome Tor Vergata, 00133 Rome, Italy; fabiana.amadori@gmail.com (F.A.); el.terracciano@gmail.com (E.T.)

**Keywords:** immunization, education, preventive health services

## Abstract

Since 2017, Italy has expanded the compulsory vaccination from 4 to 10 for those aged 0 to 16 years. Because of the great organizational effort required for the immunization services, minor attention was given to the vaccinations not included among the mandatory ones. This situation led to a real difficulty in harmonizing the vaccination procedures even inside a single region. In the Lazio region, the Laboratory of Vaccinology of the University of Rome Tor Vergata established a working group to create a new training model for healthcare professionals. The course program proposed an update of three vaccinations which are not mandatory but actively offered. It included the same part of scientific updating and a variable part based on local experiences. A specific anonymous questionnaire on knowledge and attitude was administered. The study aimed to propose a general format of training courses for vaccination centers adaptable to the individual local health units (ASLs) and to evaluate through questionnaires. The results show differences in knowledge and attitudes toward non-mandatory vaccinations among the ASLs of Lazio, confirming the usefulness of a support to make knowledge and procedures homogeneous. This model could be adapted to any healthcare setting and exported to other services.

## 1. Introduction

Healthcare is organized at a regional level in Italy. Regions are the first level of territorial subdivision of Italy, with 20 in total with populations of about 10 million to 125,000 inhabitants, and they represent a public authority with political and administrative autonomy particularly in healthcare policy [1]. The central coordination, under the responsibility of the Italian Ministry of Health, allows the publication of national documents in order to harmonize the over-regional context, making a diversified scenario as homogeneous as possible. Among these documents, the National Immunization Plan 2017–2019 (NIP) was published in February 2017 and is still in use [2]. Italian NIP identified public health priorities in accordance with the global and European vaccination goals [3,4]. among which included: reduce the burden of vaccine-preventable infectious diseases and guarantee the active and free offer of vaccinations to specific age groups and to the groups of population at risk. This document aimed to harmonize immunization strategies between the regions by providing a national strategy and specific objectives and targets [2,5].

Before the publication of the NIP, an important drop in the vaccination coverage below the 95% threshold recommended for polio and other vaccinations was reported [6,7,8]. This decline in vaccination coverage was observed in many high-income countries and was mainly due to an increase in vaccine hesitancy [7,9,10,11]. Shortly after the publication of the NIP, in July 2017, the Italian Government decided to extend by law the number of compulsory vaccines from 4 (vaccines against poliomyelitis, tetanus, diphtheria and hepatitis B) to 10 (adding vaccines against pertussis, *Haemophilus influenzae* type b, measles, mumps, rubella and varicella) for those aged 0–16 years to counteract the risk related to low vaccination coverage [12,13].

This new situation led to an organizational issue, in terms of uniformity of procedures in the way services (e.g., increase in the opening time, improvement in the vaccination booking system are provided) in the vaccination centers of the same region. In fact, immunization services were required to respond to an increased demand for compulsory vaccination in children and adolescents in a short period. This law led to positive results in terms of coverage, confirmed by a first estimation reported after just 6 months [9,14]. Because of the great effort required to respond to the law, minor attention was given to the vaccinations not included among the mandatory ones but recommended by NIP [15]. The poor consideration given to the vaccinations not made mandatory by law and the difficulty of harmonizing the procedures even inside a single region prompted the Laboratory of Vaccinology of the University of Rome Tor Vergata (LaVaUTV) to establish a working group to face this problem.

The working group decided to create a training model of course/workshop with the aim of harmonizing the work of the local health units (in Italian, azienda sanitaria locale—ASL) in the Lazio region. Lazio is a region of central Italy, its capital is Rome and there are four other provinces in the region. The number of inhabitants makes Lazio the second most populated region in Italy and the ninth by surface area. Ten ASLs, six in Rome and its province one for each of the other provinces, are responsible for the health of about 6 million inhabitants [16]. 

We decided to organize a workshop format proposing an update regarding three vaccinations which are not mandatory but whose active offer is recommended by the NIP for adolescents and adults. Pneumococcal vaccines (PCV13 + PPV23) are recommended in at risk adults and in the elderly with a sequential schedule [17,18], the anti-herpes zoster (HZ) vaccine [19] was introduced for the first time in NIP 2017-19 for subjects aged sixty-five and the recommendation for vaccine against papillomavirus (HPV) [20] was extended to male adolescents.

The aim was to propose a general format of training courses for healthcare professionals working in vaccinations centers with a homogeneous program adaptable to the resources and the epidemiological framework of the individual ASL. Moreover, we aimed to test if the courses/workshops, based on expert proposals, were well tailored and customized for each local reality while maintaining a shared format. 

## 2. Materials and Methods 

### 2.1. Organizing the Courses

The LaVaUTV organized a meeting for the working group, made up of experts in the field of vaccinations and including the vaccination coordinators for the six ASLs in Rome and its province. The six ASLs, two in the metropolitan area of the town and four in the near suburbs, cover the health needs of about 4,300,000 inhabitants [16]. The aim was offering a training course based on uniform contents adapted to the local organization and partially provided by local staff from the individual ASLs. 

During the meeting, organizational and logistic issues emerged; some of them were shared by ASL 3 and 4, and some by ASL 5 and 6, respectively. These ASLs have in common organizational and territorial characteristics, such as the number of customers served and the level of urbanization [21]. Consequently, it was agreed to organize four courses addressed to physicians and nurses involved in vaccinations to be recruited on a voluntary basis among workers in the territories of ASL 1, 2, 3–4, and 5–6, respectively. The program of the courses was tailored with contents concerning local peculiarities with the presentation of local reports and in-depth studies on specific difficulties. 

### 2.2. The Courses

For each vaccination selected, the course program included the same part of scientific updating and a variable part based on local experiences. The scientific update was held by experts, both external and internal personnel of the ASLs, and the part of local experience was held by the internal staff assigned by the coordinators of each ASL with the aim of considering specific local issues, the coverage and internal management of the vaccination program. The general program of the course is shown in Box 1.

Box 1Program of the course shared with the local health units (Azienda Sanitaria Locale, ASL).
**New Vaccines for Adolescents and Adults**
9:30Introduction: new vaccines for adolescents and adults9:40HPV, Pneumococcus, Herpes Zoster, Vaccines for adolescents and adults in ASL X 10:00HPV Vaccine10:15HPV Vaccine in ASL X: Q & A10:45The antipneumococcal vaccine in the adult at risk and in the elderly11:00The antipneumococcal vaccine in ASL X: Q & A11:30Coffee Break 12:00Anti HZ vaccine12:15Anti HZ vaccine in ASL X: Q & A12:45Vaccination strategies for adolescents and adults: guided discussion and conclusions13:30Questionnaire CME

The timing of the four half-day courses and the location was established.

A specific questionnaire on knowledge and attitude was prepared to be administered to the participants before the start of the courses.

The course was part of the Continuing Medical Education (CME) program (5.2 credits). At the end of the courses, the CME questionnaire was administered.

### 2.3. The Questionnaire

The anonymous questionnaire contained a first demographic-registry part (age, profession and ASL) and presented 14 closed-answer questions, of which five on HPV, four on pneumococcus (PNM) and four on HZ, plus one on general aspects related to training needs.

For the nine knowledge questions focused on topics contained in the NIP 2017–2019, such as infection transmission routes, vaccination schedules, co-administrations and target populations, only one correct answer was foreseen. For the four attitude questions that specifically investigated aspects related to vaccination and major difficulties encountered in proposing them, only one preference was required. Finally, the last question asked which topic was considered of greatest interest to be addressed in subsequent training courses. The topics of all 14 questions are shown in Table 1.

### 2.4. Statistical Analysis

We decided to consider as our sample all course participants who were included in the analysis. All health personnel who attended the courses were interested in and committed to participating. So, we included in the study all potentially interested to this kind of courses. The descriptive statistics of the sample were carried out stratifying by profession and by ASL, and expressed in terms of means, standard deviation of absolute number and percentage, as appropriate. 

Inferential statistics of knowledge questions were performed, with correct answers expressed in terms of percentage and the null hypothesis tested through the chi-squared (χ2) test, after the construction of contingency tables including correct, wrong, and missing answers. Statistical significance was set at *p*-values < 0.05 for questions that had a correct answer (Questions 1 to 3, 6 to 8, and 10 to 12). 

For the analysis of the attitude questions, which did not include a strictly correct answer, we represented and described data in terms of raw number and percentage of all the answers provided. For the statistical analysis, we used the chi-squared test to evaluate the differences between the various ASLs for each individual answer within each question. All data were collected in an Excel^®^ spreadsheet. We used the SPSS software v. 22.0 for all the analyses.

### 2.5. Ethics Committee

The study protocol regarding administration and evaluation of the questionnaires was approved by the Independent Ethics Committee of Fondazione PTV—Policlinico Tor Vergata, Rome with the number 140, year 2020

## 3. Results

### 3.1. Courses 

The courses were held from 22 to 30 March 2019 [21,22,23,24]. The three courses for ASL 1, 2, 3–4 personnel were all held in the same location (located in the town), while the course for ASL 5–6, personnel were held in a facility located in the territory of ASL 6. A total of 142 health professionals attended the courses.

### 3.2. Questionnaires 

We collected 136 questionnaires (83.4%). The mean age was 53.7 +/− 10.1 years; medical doctors comprised 68.4% of the cohort, and 21.3% of participants were nurses. The number of participants belonging to ASLs 1, 2, 5–6 was similar (about 20–25% each), while ASL 3–4 accounted only for about 13% of the sample (Table 2). 

Out of 136 respondents to the questionnaires, 114 (83.8%) were working in ASLs 1–6 and were considered in the following analysis. No statistical differences were found between ASLs workers and other responders.

### 3.3. Knowledge Questions 

Nine knowledge questions were analyzed (Questions 1, 2, 3, 6, 7, 8, 10, 11, 12). Missing and/or multiple answers were considered incorrect. Missing answers ranged from 1.5% to 12.5%. Multiple answers comprised 11 out of 1224 responses—0.9%, ranging from 0% to 3%. We carried out a sensitivity analysis considering a missing answer as a third answer option (therefore separately from the wrong ones). The results were robust in order to consider the missing answers assimilated into the wrong ones. The correct answers with percentages are shown in Table 3. 

Regarding the correct answers, Q1 and Q3 had the worst response rate (15.8% and 36.8%). The best was observed in Q2 and Q7 (93.0% and 80.7% respectively). The range of correct answers to the remaining questions varied from 47.4% to 70.2%. We found significant differences between ASLs for the questions Q3, Q6 and Q11. ASL 1 was better for Q3 (71.0%), ASL 3–4 for Q6 (88.9%) and ASL 5–6 for Q11 (91.7%).

### 3.4. Attitude Questions 

For the answers regarding attitudes and behaviors (Questions 4, 5, 9, and 13) (Table 4), we removed the single multiple answers (total multiple answers = 1) in Q9. For Q4, we found a considerable homogeneity in the answer “I actively recommend the vaccine” (96.5%; Range 91.7%–100%).

Regarding Q5, we found a total response rate ranging from 12.3% to 34.2% for each possible answer, with a higher number of responses regarding vaccine hesitancy, but without significant differences among ASLs. One third of the participants had not had difficulties in proposing vaccination, while about 1/3 had faced skeptical parents. 

For Q9, only 24.6% of responders preferred a PNM vaccine in the elderly to be managed by a GP, while about 2/3 gave the answer regarding the necessity of cooperation between GPs and doctors of the vaccination center. 

In Q13, we found some significant differences among ASLs. In ASL 3–4, 83.3% recommended vaccination according to NIP indication while. in ASL2, only 31.0% (*p* = 0.003); in the same ASL2, 20.7% of participants did not have a specific opinion (*p* = 0.036). 

In Q14, we evaluated 132 responses as 15 participants out of 114 (13.2%) gave more than one answer (Table 5).

In this question, almost half of the answers (45.6%) reflected an interest/need for courses about communication and 36.8% for knowledge about vaccines.

## 4. Discussion

In Italy, alignment in immunization strategies is lacking at an interregional and at a regional level [25,26,27,28,29]. The offer of health services, in particular vaccination ones, is extremely heterogeneous in some regions due to the different behaviors of the individual local health authorities. In the Lazio region, there is high diversity, and similar needs for prevention have been given very different organizational responses [30].

One of the main objectives of the NIP was to harmonize the vaccination strategies in the country. At the same time, the Ministry of Health produced documents to enable the regions to adopt in their territory the appropriate logistical and organizational measures. They were necessary to ensure an effective and efficient offer in order to achieve the vaccination coverage established by the NIP [31].

An important place among these measures is given to the information and education activities aimed at both the general population and the healthcare workers. Good knowledge has been associated with positive attitudes towards both vaccines’ efficacy and safety [32,33,34,35]. The information each individual receives regarding the disease can introduce adaptive behavior which has the potential to reduce new infections [36]. Therefore, the development of information and education initiatives is crucial to increase confidence in vaccination. Organizing courses and surveys focused on available vaccines can help in several ways. It can improve people’s knowledge and awareness about the importance of mass immunization [37], and it can allow us to understand the reasons for and against vaccination. It is therefore a starting point for planning effective vaccination campaigns [38]. In the literature, there are works that investigate knowledge and attitudes towards vaccination, mainly in parents and vaccinees [39]. There are also examples of training courses or interventions concerning vaccinations aimed at meeting the training needs of health workers and improving adherence to vaccinations by facilitating communication aspects. [29,40,41,42,43,44,45].

There has been a general increase in the number of courses providing basic education and scientific updating in the field of vaccination [44]. However, the harmonization of vaccination practices across different services is of paramount importance, even if it is necessary to respond adequately to the needs of different territorial realities. This has led the LaVaUTV to organize courses tailored to the needs of the vaccination centers of the ASLs of Rome and its province. 

One of the strengths of our course is in fact that it has been customized for selected immunization contexts and organized through the joint work of experts in the field and ASL coordinators/local staff with specific practical and organizational experience. 

Our course model provides, as another fundamental feature, the identification among trainers of one person among the internal staff within a single ASL. The course topics, while ensuring scientific and general knowledge common for all the participants of the different ASL, takes into great consideration the local problems and critical issues promoting a wide discussion among participants and in-depth analysis. All courses were directed/supervised by a member of LaVaUTV and coordinated by an ASL contact person/referent.

The questionnaire administered before the start of each course also allowed us to identify the training needs of the participants. From the answers obtained, it turned out that the choice of the topics was well centered; in fact, most of the information provided during the course was not well known and there were some discordant opinions among the participants. Through the questionnaire, we were able to receive quick feedback on the course and we were able to identify the critical issues that emerged from people who work in their specific context, knowing individual realities well. The results of the questionnaire may also be useful to modify some aspects of subsequent courses. 

Differences were found among the ASLs of the same region that should show uniform knowledge and attitudes. This result may indicate the usefulness of support to make knowledge and procedures homogeneous within the same region but also between different regions. The course also represented a moment of common reflection that involved people belonging to the same reality and allowed a broad debate; therefore, it may be defined as a workshop more than a “classic” vaccine training course.

Among the respondents to the questionnaire, 45.6% asked for more courses focusing on communication issues. It is therefore interesting to note that, in addition to specific technical aspects on the subject, communication remains a fundamental requirement in medicine and public health [46]. 

Considering that at the end of the course there was already a mandatory questionnaire provided by the CME system, we did not administer a further final questionnaire to assess the increase in knowledge. The final CME questionnaire foresees a part in which to express personal satisfaction and the results obtained were very positive.

One of the strengths of this work is the preparation of a model training course that, on the one hand, provides a scientific update common to all participants, and on the other hand takes into great consideration the problems and realities of each individual vaccination service, offering the possibility of a broad discussion and in-depth analysis among the participants. 

Moreover, the questionnaire did not only represent a tool to test the level of knowledge or opinions of participants, but also a means to obtain immediate feedback on the validity of the choices of the topics covered during the course, in order to obtain a basis for further improvements and for planning the subsequent ones.

One limitation of the study has been that the participants were healthcare workers in the vaccination field, recruited on a voluntary basis, and therefore, we would have expected the best knowledge in the proposed topics. The results, however, highlight how many questions and difficulties are still present within specific groups working on this topic.

## 5. Conclusions

Taking into account the feedback obtained from the local coordinators, the initiative has produced a level of reflection in the operators of the different vaccination centers. The acquisition of information elements in the interactive form facilitated the decision-making capacity of the operators. Other ASLs of other provinces asked the LaVaUTV to organize the same type of format. This model could be adapted to healthcare settings other than vaccination and exported to other services and realities. 

## Figures and Tables

**Table 1 vaccines-08-00514-t001:** Topics and definition of the questions.

Topic	Question	Definition
HPV	Q1	How is HPV infection transmitted?
	Q2	Which is the proposed vaccination schedule for the anti-HPV-9 vaccination?
	Q3	With which vaccination the possibility of co-administration of the HPV-9 vaccine is described in the SPC?
	Q4	Which is your usual attitude towards anti-HPV vaccination?
	Q5	Which is the main difficulty you face when proposing HPV vaccination?
PC	Q6	In which patients is PVC recommended according to NIP 2018–2019?
	Q7	Which vaccination schedule is generally used to propose PVC in the elderly?
	Q8	In addition to the flu vaccine, with which vaccines, according to SPC, can the 23-valent polysaccharide vaccine be co-administered in adults?
	Q9	In your opinion, who should administer the PVC in the elderly?
HZV	Q10	Which disease can a person with shingles transmit?
	Q11	According to the SPC, at what age live attenuated shingles vaccine can be administered?
	Q12	Which vaccinations are reported in the SPC to be co-administered with the live attenuated shingles vaccine?
	Q13	Which is your usual attitude towards Herpes Zoster vaccination?
Education	Q14	Which topic should a training course on vaccination for healthcare workers mainly deal with?

**Table 2 vaccines-08-00514-t002:** Demographic characteristics of the respondents to the questionnaires.

Responders to the Questionnaires (*n* = 136)	
Age mean (+/− SD)	57.7 (+/− 10.1)
Professions	N (%)
Medical Doctors	93 (68.4%)
Nurses	29 (21.3%)
Other	4 (2.9%)
Missing	10 (7.4%)
ASLs	N (%)
1	31 (22.8%)
2	29 (21.3%)
3	12 (8.8%)
4	6 (4.4%)
5	15 (11.0%)
6	21 (15.4%)
Other	21 (15.4%)
Missing	1 (0.7%)

**Table 3 vaccines-08-00514-t003:** Questions with correct answers.

	Correct Answers	ASL 1	ASL 2	ASL 3–4	ASL 5–6	Total	*p*-Value
		31 (27.2%)	29 (25.4%)	18 (15.8%)	36 (31.6%)	114	
Q1 (HPV)	It is also possible through contact with skin of other parts of the body and fomites	19.4%	10.3%	27.8%	11.1%	15.8%	0.32
Q2 (HPV)	Two doses in subjects aged from 9 to 14 years at the time of the first injection	90.3%	86.2%	100.0%	97.2%	93.0%	0.193
Q3 (HPV)	Diphtheria, tetanus, pertussis, poliomyelitis vaccines	71.0%	24.1%	33.3%	19.4%	36.8%	<0.0001 *
Q6 (PC)	Persons with certain risk factors	48.4%	69.0%	88.9%	50.0%	60.5%	0.015 *
Q7 (PC)	The administration of the conjugated PCV-13 followed after about 12 months by the polysaccharide PCV-23	87.1%	75.9%	83.3%	77.8%	80.7%	0.67
Q8 (PC)	No vaccine	51.6%	37.9%	38.9%	55.6%	47.4%	0.43
Q10 (HZV)	Transmit chickenpox to susceptible contacts	71.0%	44.8%	55.6%	72.2%	62.3%	0.085
Q11 (HZV)	Age ≥ 50 years	64.5%	51.7%	66.7%	91.7%	70.2%	0.004 *
Q12 (HZV)	With inactivated influenza vaccine	48.4%	58.6%	50.0%	52.8%	52.6%	0.85

* *p* < 0.05.

**Table 4 vaccines-08-00514-t004:** Attitude and behaviors questions.

	ASL 1	ASL 2	ASL 3–4	ASL 5–6	Total Answers	*p*
**Question 4—Attitudes vs. HPV**						
I actively recommend the vaccine	96.8%	100.0%	100.0%	91.7%	96.5%	ns
I do not recommend the vaccine	0.0%	0.0%	0.0%	0.0%	0.0%	ns
I’m talking about it when asked by the parents of the patients	3.2%	0.0%	0.0%	5.6%	2.6%	ns
I don’t have a clear opinion	0.0%	0.0%	0.0%	0.0%	0.0%	ns
Missing	0.0%	0.0%	0.0%	2.8%	0.9%	ns
**Question 5—Difficulty in proposing anti HPV vaccination**						
Skeptical parents	19.4%	51.7%	33.3%	33.3%	34.2%	ns
Parents safety concerns	19.4%	10.3%	5.6%	11.1%	12.3%	ns
Sexual transmission of the virus	25.8%	13.8%	5.6%	19.4%	17.5%	ns
No specific difficulties	29.0%	20.7%	44.4%	36.1%	31.6%	ns
Missing	6.5%	3.4%	11.1%	0.0%	4.4%	ns
**Question 9—Elderly anti-Pneumococcus vaccination management**						
By general practitioner	29.0%	10.3%	27.8%	30.6%	24.6%	ns
By clinician	0.0%	3.4%	0.0%	0.0%	0.9%	ns
By vaccination centre	3.2%	3.4%	0.0%	5.6%	3.5%	ns
Cooperation between general practitioner and vaccination centre	61.3%	82.8%	66.7%	61.1%	67.5%	ns
Missing	6.5%	0.0%	5.6%	2.8%	3.5%	ns
**Question 13—Attitude towards Herpes Zoster vaccine**						
I recommend it to people over 65 years of age.	51.6%	31.0%	83.3%	63.9%	55.3%	0.003 *
I recommend it to everyone with specific risk factors	22.6%	34.5%	11.1%	22.2%	23.7%	ns
I’m waiting for the inactivated vaccine to arrive	6.5%	0.0%	0.0%	0.0%	1.8%	ns
I don’t have a specific opinion	9.7%	20.7%	0.0%	2.8%	8.8%	0.036 *
Missing	9.7%	13.8%	5.6%	11.1%	10.5%	ns

* *p* < 0.05. ns: no significance.

**Table 5 vaccines-08-00514-t005:** Questions about course topics.

Question 14—Which Topic Should a Training Course Mainly Deal with?	We Show Here Only Raw Results					
	ASL 1	ASL 2	ASL 3–4	ASL 5–6	Total Answers	*p*
Communication in the field of vaccination	12 (10.5%)	16 (14.0%)	8 (7.0%)	16 (14.0%)	52 (45.6%)	ns
Knowledge about vaccines	11 (9.6%)	13 (11.4%)	8 (7.0%)	10 (8.8%)	42 (36.8%)	ns
Information about diseases that can be prevented by vaccines	1 (0.9%)	7 (6.1%)	3 (2.6%)	5 (4.4%)	16 (14.0%)	ns
Recommendations on the administration of vaccines	9 (7.9%)	7 (6.1%)	3 (2.6%)	3 (2.6%)	22(19.3%)	ns
Missing	3 (2.6%)	1 (0.9%)	1 (0.9%)	2 (1.8%)	7 (6.1%)	ns

*p* < 0.05. ns: no significance.

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
