# Peer review of "Healthcare Workers Training Courses on Vaccinations: A Flexible Format Easily Adaptable to Different Healthcare Settings"

_vaccines, 2020, doi:10.3390/vaccines8030514_

Round 1

Reviewer 1 Report

Dear Authors

I have read with great interest the manuscript titled: "Healthcare workers training courses on vaccinations: a flexible format easily adaptable to different healthcare settings ", for Vaccines journal, which I would like to make the following comments:

The study addresses the evaluation of a course on vaccines aimed at healthcare workers from an Italian region.

The abstract does not contain the objective of the study, nor results, it is confusing.

The introduction recounts the changes in vaccination policies in Italy for three years that made some vaccines in the children's schedule mandatory, after which a course for ASLs is proposed focused on three non-mandatory vaccines (Pneumococcal vaccines, Herpes Zoster and HPV males adolescents). The objective of the study is mentioned (to propose a general format of training courses for healthcare professionals) and to assess whether the courses were well adapted to the local reality.

The methodology limits itself to exposing the 4-hour program of the course, lists the questions of a test of 14 very simple items on the 3 vaccines discussed during the course and proposes a simple statistical analysis.

He says that an ethics committee has approved the course.

The course was held four times, 142 professionals attended and 136 questionnaires were collected. The results of the questionnaire responses are presented in various tables.

The discussion is irrelevant and the conclusion very daring since the proposed model has not been evaluated correctly.

The quotes in the tables and the figure are not correct. The tables are cited above and the figures below.

The evaluation of a 4-hour course with a simple test to a low sample of professionals does not contribute anything to the scientific literature.

Author Response

Dear Authors

I have read with great interest the manuscript titled: "Healthcare workers training courses on vaccinations: a flexible format easily adaptable to different healthcare settings ", for Vaccines journal, which I would like to make the following comments:

We thank the reviewer for considering our work interesting and we took in great consideration his comments

The study addresses the evaluation of a course on vaccines aimed at healthcare workers from an Italian region.

The abstract does not contain the objective of the study, nor results, it is confusing.

Thanks for the suggestion. We revised the abstract, eliminating headings and trying to make clear the objective and the results (lines 29-32)

The introduction recounts the changes in vaccination policies in Italy for three years that made some vaccines in the children's schedule mandatory, after which a course for ASLs is proposed focused on three non-mandatory vaccines (Pneumococcal vaccines, Herpes Zoster and HPV males adolescents). The objective of the study is mentioned (to propose a general format of training courses for healthcare professionals) and to assess whether the courses were well adapted to the local reality.

The methodology limits itself to exposing the 4-hour program of the course, lists the questions of a test of 14 very simple items on the 3 vaccines discussed during the course and proposes a simple statistical analysis.

The programm of the course was prepared discussing with expert and local public health authorities, after examinating the literature about courses for healthcare workers. The questions were prepared in such a way to have a simple tool for a rapid evaluation about knowledge and attitudes and we think we used the appropriate statistical analysis.

He says that an ethics committee has approved the course.

We thanks the reviewer for giving us the opportunity to explain that the Indipendent Etics Committee did not approved the course but the administration and evaluation of the questionnaire (lines 150-152)

The course was held four times, 142 professionals attended and 136 questionnaires were collected. The results of the questionnaire responses are presented in various tables.

Yes, this is what have been done

The discussion is irrelevant and the conclusion very daring since the proposed model has not been evaluated correctly.

We slightly modified the discussion following also the suggestions of the other reviewer (lines 238-240). Regarding the conclusion, we agree that we were possibly too enthusiastic and we softened the full sentence (lines 301-302)

The quotes in the tables and the figure are not correct. The tables are cited above and the figures below.

We made a check of the entire document and we corrected some small mistakes in the Tables

The evaluation of a 4-hour course with a simple test to a low sample of professionals does not contribute anything to the scientific literature.

We think we gave at least a small contribution to the topic and we hope that the reviewers, that gave us many useful suggestion, may found it.

Submission Date

22 July 2020

Date of this review

30 Jul 2020 02:13:14

Reviewer 2 Report

This paper presents unique findings related to harmonizing the vaccination procedures across vaccination centers in Italy, by proposing a format of training courses for healthcare personnel.  Here are a few recommendations:

  1. The older, six vaccines should be reported in the introduction section.
  2. The term "organizational issue", in the introduction section needs to be elaborated. Challenges related to streamlining vaccination procedures, both within and between organizations' need to be specified.
  3. What does "region" in Italy entail? Specifically, is it a congregation of cities, towns, villages? How big is a region? The authors need to clarify this in the paper.
  4. The authors need to mention the type of sample, in methods.

Author Response

This paper presents unique findings related to harmonizing the vaccination procedures across vaccination centers in Italy, by proposing a format of training courses for healthcare personnel. 

We thank the reviewer for his suggestions.

Here are a few recommendations:

  1. The older, six vaccines should be reported in the introduction section.

Thank you for suggesting this point. We described the four older compulsory vaccines and the six new (lines 56-58)

  1. The term "organizational issue", in the introduction section needs to be elaborated. Challenges related to streamlining vaccination procedures, both within and between organizations' need to be specified.

Thank you for the suggestion. We specified the most important organizational aspects (lines 59-61)

  1. What does "region" in Italy entail? Specifically, is it a congregation of cities, towns, villages? How big is a region? The authors need to clarify this in the paper.

Thank you again. We described what are Regions in Italy (lines 40-42)

  1. The authors need to mention the type of sample, in methods.

Reviewer 3 Report

In this manuscript, the authors performed a descriptive study on the knowledge and attitude of healthcare professionals upon three vaccinations not mandatory in Lazio region of Italy. They collected questionnaire data and performed some simple analysis. The topic is relevant, and method is acceptable. I have a few suggestions:

(1) The format of Abstract is not consistent with our journal. The author may need to revise it accordingly.

(2) In discussions, the authors may wish to discuss the influence of disease-behavior feedback. For example, the effect of peer pressure and interpersonal connect networks. The following two papers will be helpful to understand it: (1) Statistical physics of vaccination, Physics Reports (2016); (2) Coupled disease–behavior dynamics on complex networks: A review, Physics of life reviews (2015).

Author Response

Comments and Suggestions for Authors

In this manuscript, the authors performed a descriptive study on the knowledge and attitude of healthcare professionals upon three vaccinations not mandatory in Lazio region of Italy. They collected questionnaire data and performed some simple analysis. The topic is relevant, and method is acceptable.

We thank the reviewer for his suggestions.

I have a few suggestions:

  • The format of Abstract is not consistent with our journal. The author may need to revise it accordingly.

We revised the abstract

(2) In discussions, the authors may wish to discuss the influence of disease-behavior feedback. For example, the effect of peer pressure and interpersonal connect networks. The following two papers will be helpful to understand it: (1) Statistical physics of vaccination, Physics Reports (2016); (2) Coupled disease–behavior dynamics on complex networks: A review, Physics of life reviews (2015).

We thank the reviewer for pointing out this important aspect and we introduced this point in the discussion (lines 238-240)

Round 2

Reviewer 1 Report

Dear Authors

I have read with great interest the second version of the manuscript titled: "Healthcare workers training courses on vaccinations: a flexible format easily adaptable to different healthcare settings ", for Vaccines journal, which I would like to make the following comments:

The authors have responded to all the suggestions that were made in the previous version of the manuscript.

Tables continue to be cited below, it has not been corrected, tables are cited above and figure below.

No substantial changes have been made.

The authors have responded to the phrase "the evaluation of a 4-hour course with a simple test to a low sample of professionals does not contribute anything to the scientific literature", as: "We think we gave at least a small contribution to the topic and we hope that the reviewers, that gave us many useful suggestions, many found it ". It can be accepted as a contribution, but it is still a manuscript with a weak methodology.

The authors thank MSD Italia in the final section of Acknowledgments for their unconditional support in organizing the courses but do not indicate what it has consisted of.

Training in Vaccinology for healthcare professionals should be provided by the university or scientific societies, not by the pharmaceutical industry.

Author Response

Open Review

Comments and Suggestions for Authors

Dear Authors

I have read with great interest the second version of the manuscript titled: "Healthcare workers training courses on vaccinations: a flexible format easily adaptable to different healthcare settings ", for Vaccines journal, which I would like to make the following comments:

The authors have responded to all the suggestions that were made in the previous version of the manuscript.

  1. Tables continue to be cited below, it has not been corrected, tables are cited above and figure below.

We corrected the citation of the Tables

No substantial changes have been made.

The authors have responded to the phrase "the evaluation of a 4-hour course with a simple test to a low sample of professionals does not contribute anything to the scientific literature", as: "We think we gave at least a small contribution to the topic and we hope that the reviewers, that gave us many useful suggestions, many found it ". It can be accepted as a contribution, but it is still a manuscript with a weak methodology.

  1. The authors thank MSD Italia in the final section of Acknowledgments for their unconditional support in organizing the courses but do not indicate what it has consisted of.

We thank the reviewer and we explained that we thank MSD for the economic support without any involvement in the conceptualization, planning and evaluation of the study.

  1. Training in Vaccinology for healthcare professionals should be provided by the university or scientific societies, not by the pharmaceutical industry.

We absolutely agree with the reviewer and, in fact, our courses have been planned and organized by our University Team together with independent experts.